# Targeted Strategies for Degradation of Key Transmembrane Proteins in Cancer

**DOI:** 10.3390/biotech12030057

**Published:** 2023-09-06

**Authors:** Vehary Sakanyan, Nina Iradyan, Rodolphe Alves de Sousa

**Affiliations:** 1Faculté de Pharmacie, Université de Nantes, 44035 Nantes, France; 2ProtNeteomix, 29 rue de Provence, 44700 Orvault, France; 3Institute of Fine Organic Chemistry after A. Mnjoyan, National Academy of Sciences of the Republic of Armenia, Yerevan 0014, Armenia; nanraifok54@mail.ru; 4Faculté des Sciences Fondamentales et Biomédicales, Université Paris Descartes, UMR 8601, CBMIT, 75006 Paris, France; rodolphe.alves-de-sousa@parisdescartes.fr

**Keywords:** cancer chemotherapy, transmembrane receptors, EGFR, targeted protein degradation, furfuryl–quinolin–triazole–thiol chemicals

## Abstract

Targeted protein degradation is an attractive technology for cancer treatment due to its ability to overcome the unpredictability of the small molecule inhibitors that cause resistance mutations. In recent years, various targeted protein degradation strategies have been developed based on the ubiquitin–proteasome system in the cytoplasm or the autophagy–lysosomal system during endocytosis. In this review, we describe and compare technologies for the targeted inhibition and targeted degradation of the epidermal growth factor receptor (EGFR), one of the major proteins responsible for the onset and progression of many types of cancer. In addition, we develop an alternative strategy, called alloAUTO, based on the binding of new heterocyclic compounds to an allosteric site located in close proximity to the EGFR catalytic site. These compounds cause the targeted degradation of the transmembrane receptor, simultaneously activating both systems of protein degradation in cells. Damage to the EGFR signaling pathways promotes the inactivation of Bim sensor protein phosphorylation, which leads to the disintegration of the cytoskeleton, followed by the detachment of cancer cells from the extracellular matrix, and, ultimately, to cancer cell death. This hallmark of targeted cancer cell death suggests an advantage over other targeted protein degradation strategies, namely, the fewer cancer cells that survive mean fewer chemotherapy-resistant mutants appear.

## 1. Introduction

Advances in the development of more effective drugs and therapeutic approaches [1,2] and prevention programs to reduce the harmful effects of tobacco on smokers [3] have determined the success in reducing human cancer mortality over the past decade. Consistent with these data, the count of functional limitations among cancer survivors in the United States from 1999 to 2018 showed that their proportion increased from 57.0% to 70.1%, indicating a clear improvement in the health of cancer patients [4]. However, traditional cancer therapies have low specificity and cause serious side effects [5,6]; moreover, targeted protein inhibition causes drug resistance [7,8]. Consequently, cancer remains one of the leading causes of human death, and the treatment of various types of tumors is a huge challenge for humanity in the 21st century.

The identification and characterization of tumor-related hallmarks contributed to the development of anti-target therapeutic strategies for more than 100 types of malignancies [9,10,11]. Currently, cancer treatment includes various approaches such as surgery, radiation therapy, hormonal therapy, immunotherapy, and chemotherapy. This review focuses on cancer chemotherapy, which uses two fundamentally different strategies: standard therapy and targeted therapy. Standard chemotherapeutic agents are cytotoxic because they kill cancer cells, while targeted chemotherapeutic agents are often cytostatic because they bind to tumor cells and block cell proliferation. Approved cytotoxic drugs have relatively low tumor specificity and high toxicity, while targeted therapy increases treatment specificity and appears to reduce secondary events in patients [12]. Accurate and reliable information about the key genes and proteins responsible for the respective cancer has become the cornerstone of targeted treatment.

Targeted protein inhibition accelerated the development of new chemotherapeutic drugs capable of inhibiting key proteins in tumor cells [13]. However, it became clear that the main disadvantage of inhibitors is the occurrence of drug-resistant mutations in both the gene encoding the target protein and in other associated DNA regions, which lead to the loss of the therapeutic effect [14]. The combined use of standard chemotherapy and targeted chemotherapy was found to provide more personalized and effective treatment for cancer patients [15]. A striking example of modern therapy for aggressive cancer is intravenous treatment with platinum drugs in combination with targeted protein inhibition by chemotherapy, which improves the well-being of patients and prolongs their lives [16]. However, the inhibition of overexpressed and overactivated key proteins with small molecule inhibitors provides a limited duration of therapeutic effect on cancer.

Primary and secondary driver mutations resulting from targeted protein inhibition are often common causes of drug resistance in cancer treatment. In this review, we turn to data on the epidermal growth factor receptor (EGFR), which is one of the main tumor markers in many types of cancer. We highlight the shortcomings of the EGFR inhibition strategy in cancer treatment, leading to the need to develop an alternative strategy called targeted protein degradation (TPD). This promising strategy for the degradation of the key proteins responsible for the pathological process offers new hope for more successful cancer treatments in the future.

## 2. Targeted Inhibition of Receptor Tyrosine Kinase EGFR

Transmembrane receptor tyrosine kinases control different signaling pathways that play an important role in many cellular processes [17]. The ErbB1, ErbB2, ErbB3, and ErbB4 receptors, belonging to the ErbB family, are involved in cell proliferation, differentiation, invasion, and angiogenesis and other functions. The canonical mechanism of EGFR activation uses the EGF ligand or other growth factors that bind to the extracellular region of the receptor and lead to structural rearrangements favorable for its dimerization, which is necessary for the activation of the catalytic adenosine triphosphate (ATP)-binding site in the cytoplasmic region (Figure 1A) [18]. The subsequent phosphorylation of tyrosine residues located in the tyrosine kinase domain triggers the downstream signaling pathways required for normal cellular activity. However, the overexpression and overactivation of the EGFR in cells can provoke aberrant signaling, leading to the development and progression of cancer [19]. It should be noted that the EGFR can dimerize with other receptors in the ErbB family, as well as with other transmembrane receptors, and activate a wide range of cellular functions under the action of appropriate extracellular ligands [20].

The intracellular region of the EGFR consists of a juxtamembrane portion, a tyrosine kinase domain, and a C-terminal tail that acts as a linker for the interaction with other proteins after phosphorylation (Figure 1B) [21]. The kinase domain consists of five β-sheets and one α-helix containing the N-lobe and an α-helix containing the C-lobe, as determined to a 2.6 A° resolution. The ATP-binding domain is located between the N-lobe and C-lobe of the tyrosine kinase domain. The phosphorylation of the substrate occurs by the transfer of phosphorus from ATP to the tyrosine residue through the Asp813 residue of the catalytic loop.

The EGFR can be also activated by hydrogen peroxide (H_2_O_2_) generated during cognate ligand EGF binding to the receptor [22]. The binding of EGF to the EGFR promotes the transformation of O2 to H_2_O_2_ through the membrane-located NADPH oxidase Nox2; then, this reactive oxygen species reacts with cysteine residue at the position 797 (Cys797) in the proximity to the ATP-binding site, leading to the transition of the thiolate anion (Cys-S) to sulfenic acid (Cys-SOH), which is required for the activation of the receptor [23,24]. This cysteine is not directly required for catalysis but can be used to irreversibly block the ATP-binding site by binding to a small chemical molecule. EGF-independent EGFR auto-phosphorylation was also described in cells treated with small chemicals. In particular, the action of 4-nitrobenzoxadiazole derivatives relies on the formation of H_2_O_2_ by cytoplasmic superoxide dismutase, which is simultaneously associated with the inactivation of protein tyrosine phosphatase PTP-1B in cancer cells [25,26]. It was found that reactive H_2_O_2_ quickly binds to various proteins after the passage of lipophilic chemical agents through the cytoplasmic membrane [27]. However, the hydrogen peroxide produced by these compounds both activates and inactivates the expression of a large number of targeted proteins that ensure the functionality of various cellular processes [26,28]. Therefore, such chemical agents serving as a source of hydrogen peroxide provoke an unpredictable effect on therapeutic efficacy and cannot be considered as a suitable candidate for cancer treatment [29].

Protein degradation in mammals depends on the initiation type of autophagy, categorized as macro-autophagy, chaperone-mediated autophagy, and micro-autophagy [30]. The endocytosis of the EGFR is a micro-autophagy process, which orchestrates cellular signaling networks and can direct the fate of the receptor in cells. The ligand-bound EGFR undergoes endocytosis followed by the recycling and/or degradation of the receptor by proteolytic enzymes in lysosomes fused to endosomes [31]. Low doses of EGF activate clathrin-dependent endocytosis, which promotes sustained EGFR signaling and is the main mechanism of EGFR endocytosis in tumors in vivo [32]. High doses of ligand additionally induce clathrin-independent endocytosis, which is the main lysosomal degradation pathway for reducing EGFR signaling [33]. Overall, the endocytic degradation of the EGFR likely influences the fate of the receptor through the mechanism of ubiquitination [34].

Small chemical molecules and neutralizing antibodies were developed to inhibit proliferative phosphorylation in ErbB1- and ErbB2-mediated signaling pathways, which are considered important targets in cancer cells [35]. For more than two decades, targeting the ATP-binding site in the EGFR has been an attractive issue in medicinal chemistry. Inhibitors with reversible and irreversible mechanisms of action were developed to inhibit the catalytic site and improve patient survival compared to platinum-based chemotherapy, the former standard of care [36].

Three generations of tyrosine kinase inhibitors (TKIs) for the EGFR were developed for the treatment of cancer patients, and the respective therapeutic molecules were grouped according to their structure or the target sites in the receptors [37]. Different competitive kinase inhibitors mimic ATP and interact with the hinge region at the kinase ATP site but differ in the conformational state of the kinase they interact with [38].

According to FDA-approved inhibitors, first-generation gefitinib [39] and erlotinib [40] are reversible inhibitors of the catalytic site in the EGFR, which strongly inhibit the receptor when it is constitutively activated by in-frame deletion in exon 19 (del19) or L858R substitution of exon 21 [41,42]. These cancer-activating mutations, which occur in the tyrosine kinase domain of the EGFR, were classified as oncogenic drivers of non-small cell lung cancer (NSCLC), one of the most lethal cancers in the world [43]. Up to 70% of NSCLC patients with EGFR-mutant tumors and approximately 20% of patients receiving anti-EGFR TKIs develop brain metastases during the course of the disease [44,45]. The first-generation TKIs, gefitinib and erlotinib, initially showed a significant therapeutic response and prolonged survival in patients with non-small cell lung cancer (NSCLC). However, the secondary gatekeeper mutation T790M increased ATP-binding affinity, caused a relapse in most patients with NSCLC after 9–14 months of treatment, and led to resistance to gefitinib and erlotinib [46,47,48,49]. To overcome this resistance, a number of new EGFR inhibitors were developed that target the receptor carrying the T790M activating mutation.

Second-generation afatinib and dacomitinib are irreversible inhibitors that interact with Cys797 in the EGFR [50,51]. It is worth noting that cysteine is the most common covalent amino acid residue used to create irreversible bonds in drugs [52]. Afatinib and dacomitinib are active against exon 19 (del19) deletion and L858R mutation in exon 21, which are the most common activating EGFR mutations and oncogenes for many patients with NSCLC [53,54]. Unlike non-covalent inhibitors, covalent inhibitors block target proteins in two steps [55]. Covalent binding to target proteins first requires the formation of protein–target–inhibitor complex via non-covalent binding, which is similar to the equilibrium process used by non-covalent inhibitors. In the second rate-determining step, the electrophilic head of the covalent inhibitor is appropriately positioned adjacent to the amino acid residues at the binding site.

Third-generation osimertinib and mobocertinib are irreversible inhibitors that can overcome the acquired T790M mutation in the EGFR [56,57]. However, acquired resistance to these drugs was identified due to the C797X mutation in the EGFR, mutations in the phosphoinositide 3-kinase (PI3K) and RAS/mitogen-activated protein kinase (MAPK) pathways, ErbB2/Erb3 amplification and the mesenchymal epithelial transition factor (MET), or alterations in cell cycle genes [58]. An important intrinsic mechanism of resistance to osimertinib is a mutation in exon 20 of the EGFR, which was found to be sensitive to the inhibitor mobocertinib and the EGFR/MET bispecific antibody amivantamab. Thus, determining the mechanism of resistance to osimertinib therapy remains an important challenge in EGFR-induced cancer. Unfortunately, with regard to resistance to third-generation EGFR inhibitors, there were no major breakthroughs in cancer treatment [59]. However, clinical trials showed that the combined use of EGFR TKIs may offer new hope for the treatment of cancer. A good example is the combination therapy of lung adenocarcinoma using first- and third-generation TKIs, erlotinib with osimertinib, in patients with C797S and T790M mutations [60]. These impressive results stimulated the use of other combination therapies for resistant tumors, including platinum-based chemotherapy.

Targeting the EGFR by allosteric binding to the ATP-binding site was also described as a novel therapeutic strategy to overcome the drug resistance that emerges within this catalytic site [61]. Allosteric inhibitors were active against EGFRL858R/T790M/C797S mutants, and these compounds were proposed as fourth-generation drug candidates [61,62,63,64]. Non-allosteric molecules were also developed with activity against osimertinib-resistant NSCLC, characterized by broad molecular heterogeneity, and EGFRL858R/T790M/C797S mutant cells [65]. However, allosteric compounds can still induce drug-resistant mutations due to a similar strategy of inhibiting the EGFR ATP-binding site. Further clinical trials will allow for the selection of fourth-generation EGFR therapeutic agents.

It was stated that TKIs of all generations often show toxicity, as judged by the appearance of diarrhea, skin rashes, and other manifestations in patients. Patients undergoing cancer treatment complain of fatigue, cognitive impairment, depression, and sleep disturbances as well as diarrhea and skin rashes. These symptoms are often caused by the emergence of resistance mutations, making drugs ineffective at stopping cancer progression [66]. Regression to cancer is often the result of driver mutations in the EGFR gene itself, preventing drug binding to the ATP-binding pocket and reactivating the EGFR signaling pathways suppressed by the corresponding TKIs, which can lead to metastatic manifestations in the patient. However, according to the data of scaled DNA sequencing, a decrease in the therapeutic effect of an inhibitor is often associated with the appearance of various resistant mutations in the same tumor during the treatment of a patient [67]. This means that the resistance mutations that lead to cancer progression occur both in the EGFR gene itself and in other genes providing associated functions, which make aggressive tumors insensitive to further treatment.

Thus, the bottleneck for chemotherapy with EGFR inhibitors is the emergence of resistance mutations in the same tumor, which leads to a decrease in the effectiveness of cancer therapy. Therefore, anticancer chemotherapy clearly needs a more advanced strategy other than the inhibition of a key protein in order to overcome the limitations of the inhibition mechanism.

## 3. EGFR Degradation by PROTAC Technology

Targeted protein degradation (TPD) is a class of novel technologies based on chemically induced non-natural interactions between proteins of interest (POIs) and two major degradation systems, namely, the ubiquitin–proteasome system (UPS) and the autophagy–lysosomal pathway. Targeted protein degradation by a bifunctional small molecule, known as proteolysis-targeted chimeras (PROTACs), is one of the most exciting new technologies in medical chemistry and biotechnology. The advantage of this technology lies in its ability to attack and destroy druggable and non-druggable proteins in diseased cells and tissues, including cancer, immune disorders, viral infections, and neurodegenerative diseases [68].

TPD was first described as a heterobifunctional degrader PROTAC containing two chemical structures, one to recognize the protein of interest and the other to bind to the E3 ligase, resulting in the degradation of the targeted protein by UPS [69,70]. It was only after almost 15 years that the first catalytic small molecular degrader was developed [71], which has now become a breakthrough technology in chemical pharmacology for the creation of a new generation of TPDs capable of eliminating the proteins involved in the appearance and progression of cancer [72]. The chemical structure of a PROTAC consists of three covalently linked moieties: (i) a structure to bind to the protein of interest, (ii) a structure to recognize the E3 ligase, and (iii) a linker to conjugate the two structures, allowing the formation of the E3 ligase–degrader–POI ternary complex [73]. The mechanism of action of this ternary complex, leading to the polyubiquitination of the target protein and its subsequent degradation by the UPS, is shown in Figure 2.

UPS is a degradation mechanism of misfolded proteins in eukaryotic cells to maintain intracellular protein homeostasis [74]. In this system, proteins to be degraded are covalently labeled with 76 amino acid ubiquitin (Ub), and the tagging process is catalyzed by three enzymes: the ubiquitin-activating enzyme (E1), ubiquitin-conjugating enzyme (E2), and ubiquitin ligase (E3) [75]. Free Ub is activated by E1 and then attached to the cysteine residue of E1; Ub-tagged E1 passes its Ub to the cysteine of E2; E3 recruits the Ub-tagged E2 and E3 substrate for labeling ubiquitin at the lysine residue of the substrate. It is noteworthy that the human proteome contains more than 600 E3 ligases that provide specific recognition of substrates, two E1 proteins, and about 40 E2 proteins involved in the transfer of Ub and Ub-like proteins. Repeated ubiquitination processes generate a poly-Ub chain (linked via the Lys48 of Ub) on the target protein, which directs the substrate to the 26S proteasome for degradation [76].

A multiunit ATP-dependent protease, called the 26S proteasome, which is probably the most important protease in the cell, is involved in protein degradation by UPS, degrading proteins in the cytosol and nucleus [77]. In short, it removes the poly-Ub, unfolds the protein, and moves it into the inner chamber of the complexing protein. The synthesis of the 26S proteasome requires a significant amount of energy, and there are various mechanisms that regulate the adequate production of proteasome elements [78].

When designing the chemical structure in a PROTAC, various strategies must be considered to select a small compound that provides high affinity for the corresponding E3 ligase in recognition of the targeted protein (see Figure 2). The chemical synthesis and behavior of engineered protein degraders in diseased cells were detailed in recent reviews [79,80]. Below, we summarize the useful information on the evolution of protein degradation to eliminate the wild-type and mutant EGFR in cancer cells.

The linker plays a key role as a molecular glue in the PROTAC structure. When the linker binds to two ligands, the given protein of interest and the E3 ligase, the binding sites in both proteins can influence not only binding selectivity but also other protein degradation parameters such as microsomal stability [81,82]. Moreover, the anchor point of the linker is important for degradation efficiency. Therefore, moving the anchor point from one site to another in the generated compound can have up to a 10-fold difference in effect on the target protein, as shown in the threonine tyrosine kinase, where the value of the degradation coefficient (DC_50_) was shifted from 21.7 nM to 2.2 nM [83].

The binding strength between the warhead and the protein of interest is usually at a low nanomolar level. The structure of the ligand protein of interest can be slightly modified to provide suitable binding and optimize physicochemical properties, for example, by replacing piperidine with an N,N-diethylamino group [84] or morpholine with piperazine [85]. The choice of the right target protein is important for the development of a PROTAC, since some proteins are highly expressed in normal tissues, so their reduction can cause unpredictable toxicity. For example, ABT263-based VHLs recruiting PROTACs induce the selective degradation of BCL-XL in tumor models of T-cell acute lymphoblastic leukemia while conserving platelets due to low VHL expression [86]. It is clear that such behavior by a potential drug is unattractive for cancer treatment.

To date, 60 proteins have been developed for a PROTAC, with four types of E3 ligases including von Hippel Lindau ligases (VHL), Cereblon (CRBN), inhibitors of apoptosis proteins (IAP), and the mouse double minute 2 homologue [87]. Currently, only E3 ligases, which are typically VHL, CRBN, and IAP, are tested for PROTAC activation. Expanding the E3 ligase toolbox is important not only to degrade a wider range of proteins but also to potentially enable the more selective recognition of tissue- and organ-specific proteins. Two E3 ligases, VHL and CRBN, are widely used in the development of targeted protein degraders due to their efficiency and good expression in cancer cells [88,89].

Linker length and chemical composition are probably the most important characteristics of the targeted protein degrader structure, as they can have a strong impact on permeability, degradation, and specificity [90,91]. A shorter linker may prevent binding to the corresponding proteins due to steric hindrance. A longer linker may prevent the two ligands from being in close proximity, which is necessary for targeted ubiquitination. Different linkers based on polyethylene glycol, unsaturated alkane chains, and heterocyclic rings, due to their polarity and flexibility, were used as scaffold elements for PROTAC design [92].

Over the past seven years, PROTAC technology has been successfully applied to the degradation of numerous proteins in cancer [93,94,95]. With regard to targeting the EGFR with a PROTAC, it is important to elucidate whether this methodology could induce the degradation of transmembrane proteins, given their limited cellular localization and the questionable accessibility of membrane-bound receptors for ubiquitination via a cytosolic mechanism. In the first studies, different generations of EGFR inhibitors were used as the receptor-recognizing core to assess the effect of receptor degradation.

Two constructs based on gefitinib and afatinib were tested in cancer cells for the degradation of EGFR^del19^ and EGFR^L858R/T790M^ mutants, respectively [90]. PROTACs were shown to be able to induce the degradation of active EGFR, ErbB2, and MET receptor tyrosine kinases, including both EGFR mutants, at lower concentrations than comparable drugs that simply inhibit proteins. In control experiments using inactive diastereomeric compounds with identical physicochemical properties, degradation was shown to be enhanced compared to inhibition alone, highlighting the potential advantages of this pharmacological modality. These results strongly suggest that not only are RTKs substrates for post-translational degradation but also the signaling inactivation and growth inhibition achieved by protein degraders are more potent and less susceptible to kinome rewiring than those achieved through RTK inhibition. Unlike EGFR inhibitors, the new degraders act on the entire targeted protein, resulting in the blocking of all receptor functions. This study was instrumental in targeting the degradation of the EGFR and other RTKs. Moreover, a PROTAC has become something of a semolina for chemists and biologists interested in expanding the technological possibilities of the development of new therapeutics against various diseases, especially against cancer [96].

It was shown that small molecular degraders based on allosteric EGFR-TKI selectively inhibit the proliferation of cancer cells carrying the EGFR^L858R/T790M^ double mutant and the osimertinib-resistant triple mutants, EGFR^L858R/T790M/C797S^ and EGFR^L858R/T790M/L718Q^ [64]. Unfortunately, these compounds were inactive at the wild-type receptor. However, given their high antiproliferative activity, allosteric compounds remain promising candidates for the development of suitable degraders of clinically relevant EGFR mutants.

Gefitinib-based PROTAC molecules were shown to enhance degradation activity against mutant EGFR^del19^ during cell starvation [97]. These compounds strongly induced the degradation of the mutant but not the wild-type EGFR in cancer cell lines and effectively suppressed the growth of lung cancer cells. Serum starvation was found to enhance the degradation effect, which may lead to improved efficacy in tumors where serum deprivation is often observed. Overall proteomic analysis showed that the engineered compounds have high selectivity for the EGFR. In pharmacokinetic studies in mice, one of the degraders, which recruits VHL E3 ligase, showed increased activity.

Osimertinib-based PROTAC compounds were synthesized and evaluated for cytotoxicity against the NSCLC cell line. The selected compounds had a high IC_50_ value of 0.413 μM in PC9 (EGFR^del19^) cells and 0.657 μM in H1975 (EGFR^L858R/T790M^) cells [98]. The tested compounds arrested the growth of mutant cells in the G0/G1 phase by the induction of apoptosis. Carnertinib-based covalent compounds were also reported to cause the high-level degradation of the EGFR^del19^ mutant and antiproliferative activity against cancer cells but the moderate degradation of EGFR^L858R/T790M^ and weak antiproliferative activity against these double-mutant-harboring cells [99]. In this study, the role of the autophagy–lysosomal pathway in EGFR degradation appeared to be more important than expected. A number of degraders were also designed and developed by linking a previously selected pyrido-[2,3-d]pyrimidin-7-one-based compound to the appropriate ligands and using linkers of various lengths to degrade the EGFR^L858R/T790M^ double mutant in cancer cells [100]. One of the potent compounds selectively degraded the mutant receptor with a DC50 value of 5.9 nM but had no apparent effect on the wild-type EGFR. This study confirmed that the VHP ligase pathway provides the highest protein degradation among the tested ligases.

Another research group developed PROTACs inducing the degradation of the EGFR^del19^ and EGFR^L858R/T790M^ mutants in HCC827 and H1975 cells, respectively [101]. The use of starvation medium without fetal bovine serum enhanced EGFR degradation. An increased amount of the autophagy biomarker protein LC3α compared to its precursor LC3β [102] indicated the autophagic degradation of the mutant EGFR, possibly via the autophagy–lysosomal system [103]. As shown later, the covalent binding strategy proved to be an effective approach in the development of degraders based on pyrimidine and purine structures [104]. However, a compound containing a pyrimidine fragment exhibited the moderate degradation of the EGFR mutants, which prompted the authors to switch to the synthesis of purine-containing covalent degraders. One of the purine-based agents significantly inhibited the cell colony formation and growth of the H1975 and HCC827 cell lines at concentrations as low as 30 nM and 1 nM, respectively, but was much weaker against cells harboring the EGFR triple mutants. This compound remarkably reduced the phosphorylation level of the EGFR, ERK, and AKT in H1975 and HCC827 cells at concentrations as low as 100 nM and 3 nM, respectively, while leaving phosphorylated B-RAF cells intact. Furthermore, the DC_50_ values of the EGFR^L858R/T790M^ and EGFR^del19^ mutants were 1.56 nm and 0.49 nM, respectively. After the complete consumption of new compounds, the content of the mutant EGFR^L858R/T790M^ began to rapidly recover.

It should be noted that acquired clinical resistance to EGFR inhibitors in cancer is usually the result of a tertiary mutation combining the C797S mutation, MET amplification, and KRAS mutations [105], among which the C797S mutation in the EGFR is the dominant one [106]. New degraders were synthesized to eliminate the receptor upon the expression of the C797S mutation in EGFR triple-mutant cells [107]. One of the allosteric compound-based degraders showed advantage in eliminating the EGFR^L858R/T790M/C797S^ triple mutant from cancer cells while the degradation of another important triple mutant, EGFR^del19/T790M/C797S^, was unsuccessful.

PROTACs provide the cyclic ubiquitination of the targeted protein, resulting in a decrease in the actual number of active degraders capable of damaging the target protein in diseased cells. The technology is not tissue-specific and readily distributes in non-target tissues after systemic administration, causing protein dysfunction in normal tissues [108]. Additionally, the side effects caused by off-target toxicity severely limit the continued use of targeted protein degradation [109,110]. Using antibody conjugates, degraders can be specifically delivered to the cancer-associated target cells [111]. To overcome these limitations, pro-PROTACs were constructed by introducing endogenous and exogenous stimulus-responsive cellular motifs into the key binding sites of hetero-bifunctional molecules [112,113]. A non-invasive exogenous stimulus was also applied to create photo-guided drug delivery using light- and near-infrared-activated PROTAC platforms [114,115,116].

A significant disadvantage of chemotherapy is associated with its negative effect on the immune system, while alternative immunotherapy with monoclonal antibodies (mAbs) has a positive effect on the immune system [117]. Various mAbs targeting the extracellular region of the EGFR were developed and approved by many laboratories to inhibit receptor activity in cancer. Among them, cetuximab, matuzumab, panitumumab, and necitumumab were used to treat patients with NSCLC [118,119]. These anti-EGFR mAbs bind to the extracellular region of the receptor, block the ATP-binding site, and can result in the inhibition of EGFR activity [120].

To date, more than a dozen protein-degrading drugs have entered phase I clinical trials, and two therapeutic agents are in phase II trials for the treatment of ER^+^/HER2^−^ breast cancer and castration-resistant prostate cancer [121,122,123]. However, the search for the chemical and immunological inhibitors of the EGFR in phase I and II clinical trials continues with the hope of discovering reliable therapeutic drugs [124].

## 4. Alternative Strategies for EGFR Degradation

Inspired by the attractiveness and success of small molecule disruptors, novel antibody-based degraders such as the lysosome-targeting chimera (LYTAC) [125], autophagosome-binding compounds (ATTEC) [126], the autophagy-targeting chimera (AUTAC) [127], the AUTOphagy targeting chimera (AUTOTAC) [128], and specific and non-genetic IAP-dependent protein erasers (SNIPERs) [129] were also developed for cancer treatment. These approaches are based on the use of the autophagy–lysosomal degradation system, which is independent of the proteasomal degradation system and ensures the degradation of misfolded proteins and damaged organelles through endocytosis, phagocytosis, and autophagy in cells [130,131,132,133]. The activity of such protein degraders in cancer cells is of great importance for the development of new drugs for the treatment of oncological, neurodegenerative, and other diseases.

A special place among these approaches is occupied by LYTACs technology, which is designed to degrade extracellular proteins, including cell surface receptors, membrane proteins, and secreted proteins via the lysosomal pathway [125]. To implement this approach, the cation-independent mannose-6-phosphate receptor (CI-M6PR) is covalently conjugated with the corresponding monoclonal antibody to recognize the target protein located in the cytoplasmic membrane (Figure 3). The complex is then taken up by endocytosis and cleaved through the autophagic-lysosomal pathway. Cetuximab, an FDA-approved anti-EGFR antibody, resulted in the 70% degradation of the EGFR in HeLa cells. EGFR degradation by GalNAc–LYTAC attenuated EGFR signaling compared to antibody inhibition. LYTAC, containing a 3.4 kDa binding peptide coupled to a tri-GalNAc ligand, was shown to degrade integrins and reduce cancer cell proliferation. LYTAC methodology not only shows the high degradation of target proteins but also provides selectivity for cell types [130,134]. These impressive results under the leadership of Bertozzi C. R. (Nobel Prize in Chemistry 2022), in the field of developing a new technology for the targeted degradation of extracellular and membrane proteins, are of great importance for expanding the range of targeted proteins in the treatment of cancer.

Despite significant progress in technological diversity, targeted protein chemotherapy does not lead to the death of cancer cells. It can be assumed that TPD is rather useful in slowing down the progression of cancer but not in completely stopping the development and progression of the tumor. Therefore, the search for other therapeutic strategies for the degradation and elimination of key cancer proteins is an important therapeutic challenge.

## 5. New Allosteric Chemicals Bind to EGFR and Lead to Cancer Cell Death

In 2019, we described a targeted protein degradation based on the use of allosteric small molecules that can degrade a transmembrane receptor and lead to the death of cancer cells [135]. The elucidation of the molecular processes involved allowed us to explain the mechanism of targeted EGFR degradation and postulate an alternative vision for cancer chemotherapy.

We designed and synthesized furfuryl derivatives of 4^_^allyl^_^5^_^[2^_^(4′-alkoxyphenyl)^_^quinolin^_^4^_^yl]^_^4H^_^1,2,4^_^triazole^_^3^_^thiol (FQTT), by combining three scaffolds into one molecule (Figure 4A). Alkyl ester substituents of various lengths were attached to the benzene ring to obtain modified compounds capable of increasing the sensitivity of the target protein to the action of proteases. Western blotting showed a significant reduction in the EGFR and tyrosine phosphorylation bands in breast cancer, prostate cancer, and lung cancer cells treated with FQTT. In addition, a decrease in the amount of some proteins not associated with the EGFR signaling pathways was observed, suggesting the involvement of protein degradation [134,135]. MDA MB-468 triple-negative breast cancer cells lack expression of the estrogen receptor, progesterone receptor, and ErbB2 but overexpress the EGFR and tend to metastasize to major organs [136]. MDA MB-468 cells were used to investigate the reason for the decrease in EGFR levels caused by new chemicals. Notably, the cytotoxicity of the most active compounds, VM26 and VM25, at an IC_50_ of 12.6 μM and 14.4 μM, respectively, is similar to that of gefitinib, a well-known anti-EGFR drug in cancer therapy, at an IC_50_ of 15.2 μM.

FQTT compounds bind to an allosteric pocket located in close proximity to the EGFR catalytic site and cause only a short-term and very weak inhibition of the EGFR tyrosine phosphorylation in cells in the presence of the EGF ligand. Molecular dynamics modeling showed that a short chain like methyl or ethyl in the compounds is not bulky enough to fill the hydrophobic allosteric pocket, while a longer chain like butyl or pentyl almost completely occupies this space (Figure 1C). This can lead to the interaction of long-chain alkyl ethers with the Met766 residue located in the αCβC loop near the ATP-binding site and cause the possible degradation of the EGFR.

A distinct increase in the amount of the autophagy biomarker LC3β was revealed due to the transition from LC3α to LC3β in cancer cells treated with FQTT derivatives [135]. The transition from LC3α to LC3β indicates an increase in the activity of the autophagosomes responsible for the endocytic degradation of the EGFR, as previously proven by other authors [137]. Therefore, the significant degradation of the transmembrane receptor by small allosteric chemicals observed in our study (Figure 4B), initiated by lysosomes during endocytosis, was called allosteric autophagy (alloAUTO).

Heat-shock chaperones (Hsp) bind to more than 700 misfolded proteins and protect them from the ubiquitination and subsequent degradation by the 26S proteasome during the proliferation, invasion, metastasis, and death of cancer cells [138]. The Hsp90a chaperone is highly expressed in cancer cells, and suppression of the defense mechanism leads to the degradation of misfolded client proteins by cellular proteasomes [139]. Interestingly, Hsp90 recognizes the αCβ4 loop in the intracellular domain of the EGFR tyrosine kinase and likely binds to it, resulting in the blocking of the catalytic site [140,141]. We demonstrated that EGFR degradation by FQTT derivatives is associated with a decrease in the amount of Hsp90α detected by immunoprecipitation with anti-EGFR mAb [142]. This means that a decrease in EGFR folding should promote the greater degradation of the receptor protein in addition to the degradation of the EGFR caused by the autophagy–lysosomal system in endosomes.

The incubation of cancer cells for more than two hours with FQTT derivatives leads to the significant degradation of the EGFR, which is accompanied by a decrease in the content of cytoskeletal proteins β-actin and α-tubulin, commonly used as load controls in Western blotting. This unexpected effect was caused by the detachment of cells from the surface of immunological wells. The action of each compound upon cell detachment was quantified by counting the quantities of the different proteins in the cell culture supernatant collected over adherent cells after treatment with the respective compound.

Traces of the EGFR were detected in cells attached to the extracellular matrix (ECM), while the EGFR almost completely disappeared from the detached cells, especially in serum-free media compared to serum-supplemented media after exposure to FQTT derivatives. It was suggested that, in addition to the degradation of the EGFR, these chemical compounds cause a strong detachment of cancer cells from the extracellular matrix, especially under conditions of cell starvation for EGF or glutamine.

Nutrient deficiency is known to disrupt the EGFR-driven signaling cascade, leading to cell detachment from the ECM and, ultimately, programmed cell death, known as anoikis (a Greek word meaning “loss of home”) [143]. The metastatic progression of inflammatory tumors suggests that cancer cells are more resistant to anoikis compared to healthy cells. The cytoskeleton consists of actin polymers and microtubules formed by tubulin polymers, which, together with other proteins, allow integrins to attach to the ECM [144]. The phosphorylation status of the EGFR in downstream signaling pathways determines the functional state of integrins, which are transmembrane receptors that mediate cell adhesion to the ECM. Importantly, the EGFR regulates normal cytoskeletal function via the MAPK/ERK pathway by the phosphorylation of the proapoptotic Bim, a sensor protein important for interaction with microtubules [145]. Interruption of this signaling pathway by blocking Bim phosphorylation leads to the sequestration of the cytoskeleton and the detachment of healthy cells.

Defective regulation of apoptosis is considered one of the hallmarks of cancer [10]. The apoptosis pathway is controlled by the Bcl-2 protein family, which contains both pro-apoptotic and pro-survival members that balance the decision between cell life and death. The Bcl-2 family of proteins plays a critical role in apoptosis initiated by internal signaling by regulating the integrity of the mitochondrial outer membrane (MOM) [146]. The study of the dynamic interaction between the proteins of the BCL-2 family and how they control the apoptotic death of healthy and diseased cells remains an important task in the treatment of malignant tumors. Three isoforms of Bim result from alternative mRNA splicing and give rise to the Bim_EL_, Bim_L_, and Bim_S_ proteins [147]. All three Bim isoforms contain a BH3 domain required for binding other Bcl-2 family proteins and a C-terminal sequence for binding to the MOM protein [148]. The overexpression of the pro-apoptotic Bim protein was found in the cytoskeleton and mitochondria, and this protein is constitutively overexpressed in prostate and breast cancer cells as well as in primary tumor cells [149]. Recent studies on the dynamic interactions among BCL-2 family proteins and how they control the apoptotic death of healthy and diseased cells opened up new avenues for therapeutic intervention [150]. How Bim is activated in different types of cancer cells is still unknown.

To understand whether the destabilization of the cytoskeletal mechanism is related to the status of the Bim sensor protein, protein expression was assessed using immunofluorescence imaging and Western blotting. Kinetic analyses showed a transient and significant increase in Bim_EL_ expression after one hour of exposure of cells to FQTT compounds in serum-deprived medium, followed by a decrease in the level of this protein after three hours of exposure (Figure 4C). Notably, high level of Bim_EL_ expression was associated with a decrease in EGFR expression after one hour. The amounts of lysosomal protease LAMP-2 and cytoskeletal protein β-actin decreased later compared to the EGFR. This two-rate decrease in the abundance of functionally unrelated proteins (Figure 5A) appears to reflect two processes: the early and rapid degradation of the EGFR by endocytosis, followed by the slower degradation and disintegration of the cytoskeleton due to Bim sequestration.

To find out which major nutritional factors are involved in Bim-induced sequestration, protein profiles were compared in serum-deprived cultures after the addition of EGF, glutamine, or both for 6 h. The addition of glutamine or a mixture of glutamine with EGF, and, to a lesser extent, the addition of EGF alone, increased Bim_EL_ expression compared to vehicle (Figure 5B). In addition, a significant increase in the rate of Bim_EL_ Ser69 phosphorylation was found, which is likely due to an earlier increase in tyrosine phosphorylation in the EGFR at Tyr1068 leading to the activation of the signaling pathways. Thus, the replenishment of the medium with a fresh portion of glutamine improved the functional state of the Bim protein in cancer cells not exposed to FQTT derivatives.

Serum-deprived cells exposed to **VM26** after the addition of EGF, glutamine, or both showed slightly increased levels of expression of the EGFR, LAMP-2, β-actin, and probably cleaved caspase 3, compared with the low expression of these proteins in cells exposed to only **VM26** [135]. BimEL expression was slightly reduced after EGF, glutamine, or both were added to a **VM26**-treated culture, while this protein was more expressed in an untreated culture after nutrient addition. Meanwhile, nutrient supplementation increased BimEL phosphorylation at Ser69 in a culture not exposed to **VM26** but strongly suppressed Bim phosphorylation, regardless of the addition of EGF, glutamine, or both to a culture exposed to **VM26**. It becomes clear that the suppression of Bim_EL_ phosphorylation is associated with the blocking of EGFR phosphorylation required for Bim activation. These data demonstrate the ability of the allosteric degraders of the EGFR to influence the metabolic and energy balance in glutamine-deprived cancer cells.

Why does glutamine deficiency increase the ability of FQTT compounds to kill cancer cells? First, cancer cells rapidly grow and require more energy for protein synthesis than normal cells. Second, glutamine is converted to α-ketoglutarate, which feeds the tricarboxylic acid cycle with more ATP. Third, the EGFR uses ATP to activate the catalytic site and to phosphorylate more than 30 amino acids, including 7 tyrosine residues, when the receptor undergoes endocytosis [151]. Fourth, glutamine is not stable at 37 °C, and the addition of glutamine to the culture medium is always necessary. Fifth, the ATP-dependent activity of a large amount of the Hsp90α chaperone is required to correct many misfolded proteins and protect them from the degradation of the 26S proteasome [139]. Therefore, the dependence of EGFR activation and the associated signaling processes on the content of glutamine in cells can be formulated simply as “no glutamine, no EGFR signaling”.

## 6. New Vision on Cancer Chemotherapy

Cell death is a fundamental biological function that completes various processes in human life, such as embryonic development or the removal of damaged cells in diseases. Based on the molecular mechanism of action of the factors that stimulate death, biological death is represented by two main categories: programmed cell death and unprogrammed cell death [152,153]. According to this classification, unprogrammed cell death includes 1 type of cell suicide, namely, necrosis, while 12 other types of cell death are recognized as programmed non-apoptotic cell death [154]. Combining different scientific views on the relationship between oncogenesis and programmed cell death is an important issue for the development of effective therapeutic agents against cancer. In this regard, anoikis is a type of programmed apoptotic cell death that behaves differently than programmed non-apoptotic cell death. Signaling pathways triggered by the EGFR protect normal and cancer cells from anoikis [155,156].

FQTT compounds bind to a hydrophobic allosteric pocket located in the immediate vicinity of the ATP-binding site in the EGFR (see Figure 1C). An important role in this binding seems to be played by the reorientation of chemicals from Arg803 to Arg841, which is consistent with the participation of Arg841 in the dynamic changes preceding the sulfenylation of Cys797 [135,157]. This probably leads to the interaction of the longer alkyl ether chains of compounds with Met766 in the aCb4 loop located near the ATP-binding site. This rearrangement may accelerate and/or enhance the endocytic degradation of the EGFR. Induced EGFR depletion leads to the sequestration of Bim, which provokes the breakdown of the cytoskeleton. Two different authentic pathways, endocytic and cytoplasmic degradation, promote cell detachment. This course of logically connected events reflects the functional interplay that precedes the death of cancer cells.

The effect of EGFR degradation by FQTT compounds has fundamentally different consequences for cells than the inactivation of tyrosine kinase activity by inhibitors [158,159]. Targeting EGFR degradation has an advantage over EGFR inhibition because it promotes a more specific interruption of Bim phosphorylation, leading to the death of cancer cells. Unlike cytostatic TKIs against the EGFR, the allosteric degraders of the EGFR affect cell survival rather than growth and induce cancer cell death like cytotoxic molecules. This unexpected biological scenario is reminiscent of the return of “immortal” cancer cells to programmed cell death, anoikis. This means that EGFR allosteric inhibitors are not “cancer cell killers” but are, rather, molecules that restore the lost ability of cancer cells to die like normal cells after a limited number of generations. Notably, cancer cell death resulting in a tumor size reduction by allosteric FQTT compounds was confirmed in vivo in a mouse model of sarcoma [160].

The proposed mechanism of targeted protein degradation suggests that the allosteric degraders of the EGFR are promising agents for the chemotherapy of human metastatic tumors (Figure 6). Shutting down phosphorylation pathways by potent TKIs in proliferating cancer cells creates selective conditions for the emergence of different mutants through alternative mechanisms, such as H_2_O_2_ release, in the branched EGFR interactome in the tumor microenvironment. Conversely, protein degradation due to EGFR depletion results in cancer cell death (Figure 6A), leaving fewer cells to proliferate and reducing the chance of new, resistant mutations emerging. We believe that this study opens up opportunities to attenuate the metastatic progression and reduce the drug resistance in malignant tumors associated with the aberrant behavior of the transmembrane receptors in cancer cells.

Thus, our data highlight the importance of the alloAUTO strategy in targeting protein degradation through the autophagy–lysosomal and ubiquitin–proteasome pathways. The autophagy response provides the allosteric degradation of the EGFR, leading to a decrease in the activity of the downstream signaling pathways, followed by disabling the phosphorylation of the Bim sensor protein and the destruction of the cellular cytoskeleton, which, ultimately, leads to the death of cancer cells (Figure 6B). The dual mechanism probably reflects the benefits of the cytotoxic effect of standard chemotherapy and the cytostatic effect of targeted chemotherapy on cancer cells, namely, the emergence of fewer resistant mutants in tumor treatment.

## 7. Conclusions

The targeted degradation of the receptor tyrosine kinase by PROTAC chemotherapeutic agents has a number of disadvantages, as described on the Creative Biolabs PROTAC website [161]. Apparently, the main disadvantage of a PROTAC is that it does not provide cell specificity with respect to the degradation of the mutant protein, which may become a limiting factor for the use of the developed molecule as an anticancer drug in clinical trials. Solving this issue requires the use of an additional approach, which should ensure the specificity of recognition of the targeted protein in the desired cells. LYTAC likely solves this problem by using a conjugated chemical molecule with a specific antibody, which allows the mixed immunochemical agent to recognize and then destroy the targeted protein in the appropriate organ and tissue of the patient.

The treatment of tumors caused by transmembrane receptors requires a better understanding of the molecular mechanisms to improve the technological reliability of the degradation of the targeted proteins. Our data on cancer cell death during the degradation of the EGFR by chemical compounds are also of interest for the development of other transmembrane proteins’ degradation. In particular, the evaluation of compounds derived from FQTT in preventing the progression of tumors caused by an EGFR-paired receptor can be important for further research. Cancer cell death by the alloAUTO strategy look likes pro-apoptotic anoikis, which does not rule out other pathways leading to cancer cell death under the action of other agents. Elucidating this question could lead to new, more attractive strategies for the simultaneous degradation of proteins and the destruction of cancer cells in order to prevent the emergence of resistant mutants. Therefore, the clear advantages of the technologies aimed at the degradation of the key transmembrane proteins in tumor cells maintain hope for improved cancer treatment.

## Figures and Tables

**Figure 1 biotech-12-00057-f001:**
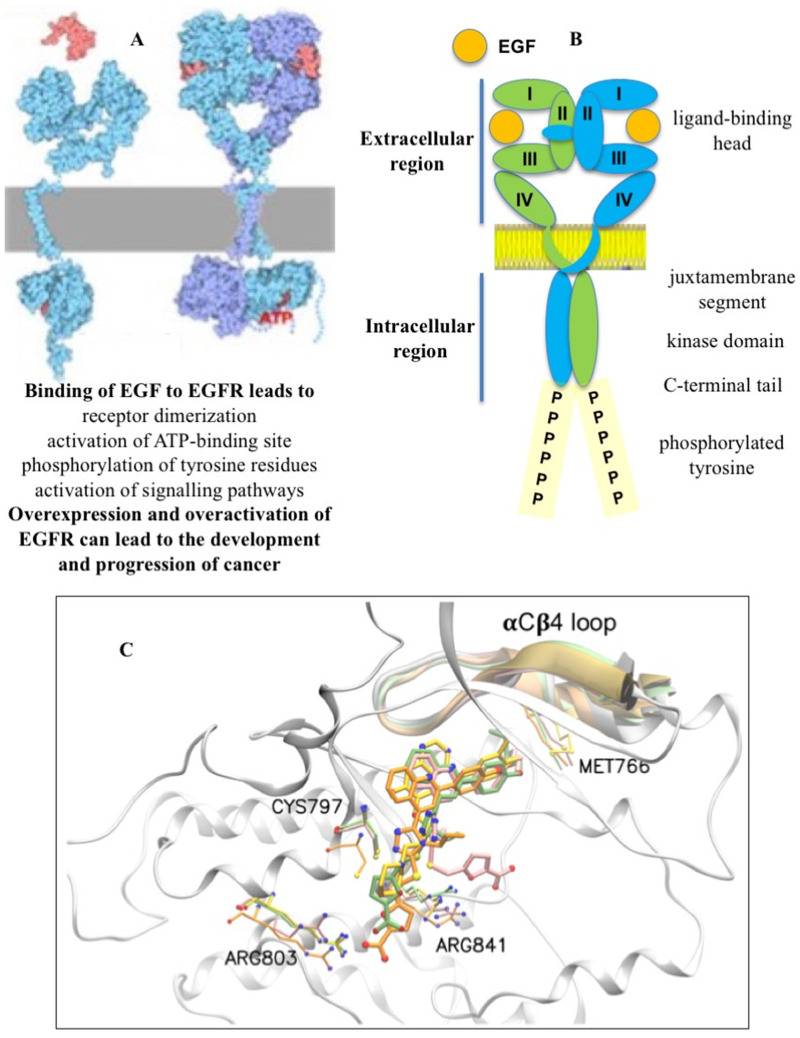
EGFR dimerization (**A**), structural features of the dimeric form of EGFR (**B**), and binding of FQTT compounds **VM3** (orange), **VM25** (pink), **VM26** (green), and gefitinib (yellow) to EGFR (**C**).

**Figure 2 biotech-12-00057-f002:**
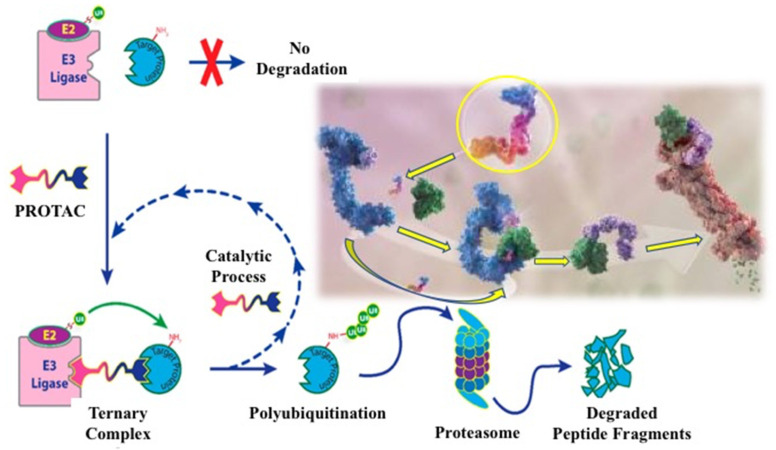
Development process for a heterobifunctional PROTAC compound designed to degrade a target protein. Molecular image of a small degrader (circled in yellow) showing its binding to ubiquitin ligase (blue) and targeted protein (green) [73].

**Figure 3 biotech-12-00057-f003:**
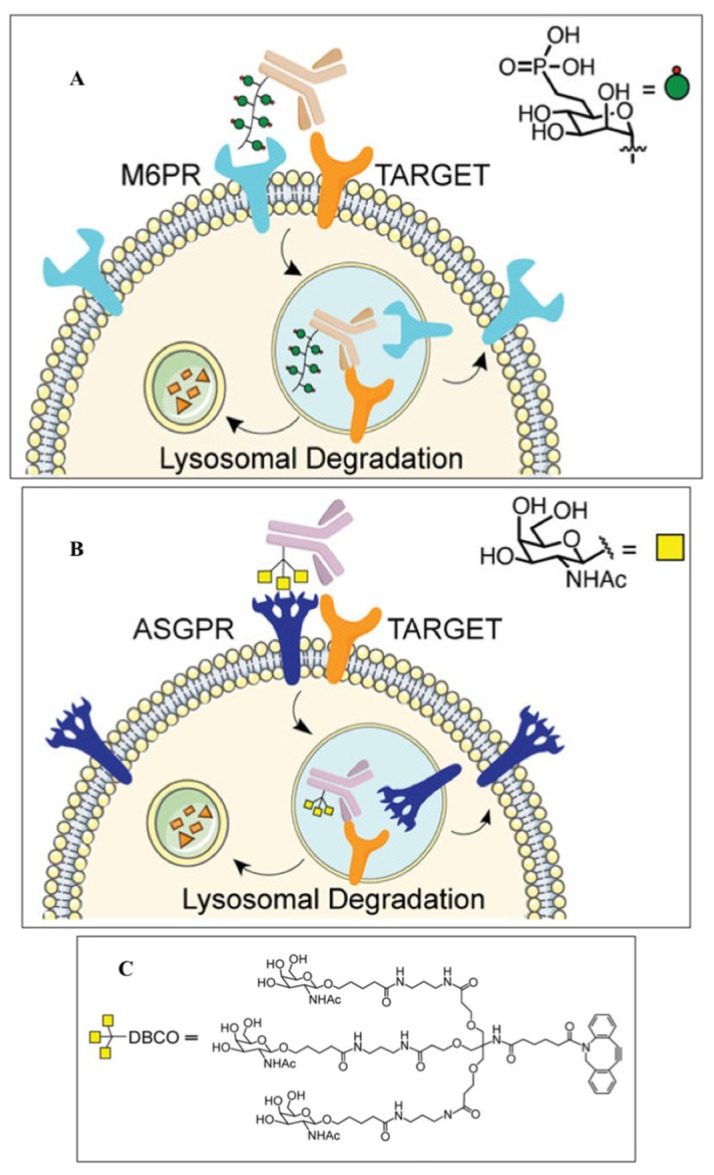
Functionality of LYTAC for targeted and cell-specific degradation of proteins [134]. (**A**) First-generation LYTACs use the cation-independent mannose-6-phosphate receptor (CI-M6PR) to degrade EGFR; (**B**) GalNAc-LYTAC captures the liver-specific asialoglycoprotein receptor (ASPGR) to specifically target hepatocytes; (**C**) tri-GalNAc-DBCO ligand structure for ASGPR targeting.

**Figure 4 biotech-12-00057-f004:**
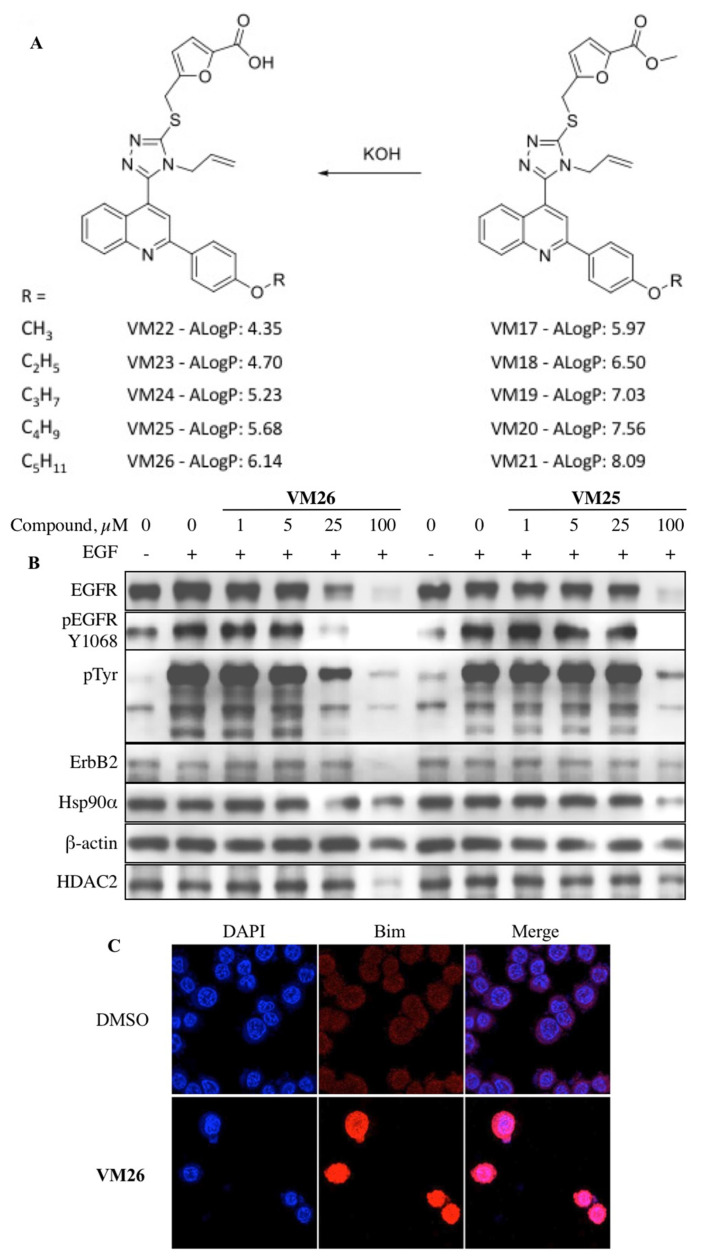
Targeted protein degradation in cancer cells with heterocyclic compounds [135]. (**A**) Furfuryl derivatives of 4^_^allyl^_^5^_^[2^_^(4′-alkoxyphenyl)^_^quinolin^_^4^_^yl]^_^4H^_^1,2,4^_^triazole^_^3^_^thiol (FQTT); (**B**) compounds **VM26** and **VM25**’s dose-dependent degradation of EGFR and other proteins and reduced phosphorylation of EGFR; (**C**) Bim response to the compound VM26 detected by immunofluorescence imaging.

**Figure 5 biotech-12-00057-f005:**
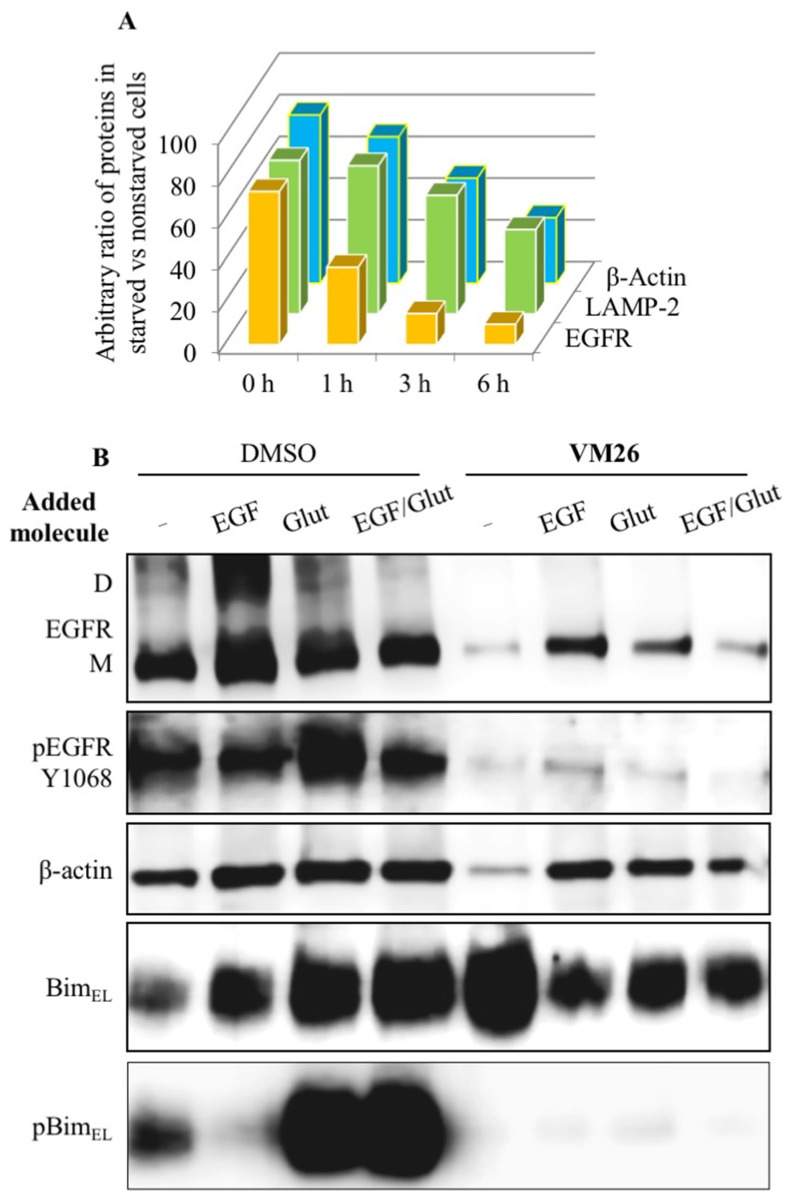
Sequestration of Bim in cancer cells by allosteric degraders of EGFR [135]. (**A**) Two-step protein degradation in starved cells vs. non-starved cells, considering 100% of each protein in non-starved cells; (**B**) impact of EGF and glutamine on protein expression and phosphorylation in untreated cells and cells treated with **VM26** in serum-deprived medium.

**Figure 6 biotech-12-00057-f006:**
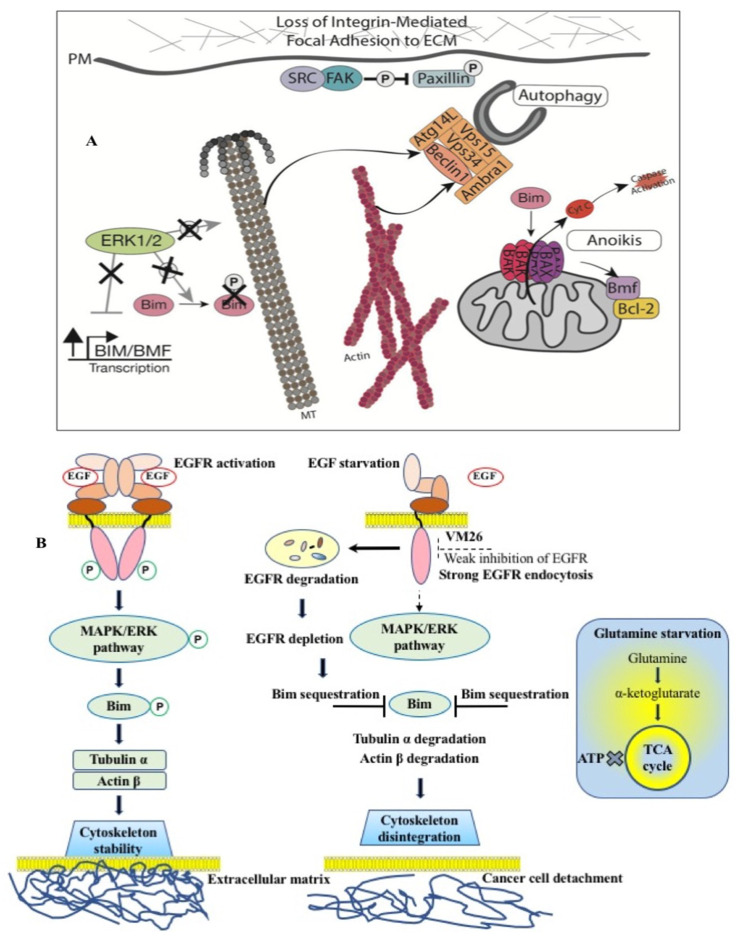
Mechanism of action of EGFR-specific allosteric degraders in cancer cells. (**A**) The role of Bim protein phosphorylation in connecting cells to the extracellular matrix, which ensures cell stability and cell proliferation [144]. (**B**) FQTT compounds bind to a allosteric site in EGFR, inducing degradation of the receptor in endosomes. Depletion of EGFR leads to sequestration of Bim, followed by disintegration of the cytoskeleton and detachment of cancer cells from the extracellular matrix. Glutamine starvation causes a deficiency of α-ketoglutarate and an inability of cells to replenish the tricarboxylic acid (TCA) cycle and produce ATP. Double starvation of EGF and glutamine reinforces cytoskeleton disintegration leading to cancer cell detachment-promoted death [135,142].

## Data Availability

The Data are available upon request from the authors.

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
