# Peer review of "Targeted Strategies for Degradation of Key Transmembrane Proteins in Cancer"

_biotech, 2023, doi:10.3390/biotech12030057_

Round 1

Reviewer 1 Report

The authors of the paper titled "Targeted Strategies for Degradation of Key Transmembrane Proteins in Cancer" have done a commendable job in thoroughly reviewing current investigations into the strategic degradation of targeted proteins, specifically EGFR, for cancer treatment. The review is wide-ranging and illuminating. Nonetheless, there are some points that could use further attention:

1. The inclusion of a table to encapsulate and contrast the various strategies would greatly enhance the understanding of the subject.

2. There is room for further discourse on the potential hurdles that targeted protein degradation might encounter, along with the future trajectory for its development.

3. In Line 959, there are two "134"

Author Response

To Reviewer 1.

Dear Reviewer, first of all, thank you very much for appreciating our review.

Please find below our answers to your suggestions.

  1. The inclusion of a table to encapsulate and contrast the various strategies would greatly enhance the understanding of the subject.

Response: It is not easy to present such a table when the topic of targeted protein degradation is rapidly developing and requires constant attention. Therefore, we invite the reader to take a look at the Creative Biolabs® website, which describes the advantages and disadvantages of the flagship PROTAC technology (lines 639-640, ref. 161 highlighted in red in chapter 7. Conclusions). We do not think that copying these explanations in our review is more necessary. In addition, the text mentions some disadvantages of targeted inhibition and degradation of proteins.

  1. There is room for further discourse on the potential hurdles that targeted protein degradation might encounter, along with the future trajectory for its development.

Response: We agree with you. Therefore, we highlight the most important disadvantage of PROTAC compared to LYTAC technology in terms of cell-specific action, where targeting to the catalytic site seems unnecessary. I think a brief explanation (lines 640-647) along with the data described in the text should suffice. In addition, we confirm the advantages of our technology in targeting protein degradation and cancer cell death to reduce the emergence of resistant mutants (lanes 629–636, highlighted in red in chapter 6. New vision on cancer chemotherapy), as well as its importance in targeting other proteins in cancer cells (lanes 28–29).

  1. In Line 959, there are two "134"

Response: Thank you, it is deleted.

Reviewer 2 Report

the research focus on a promising point that is needed in the field

Author Response

Thank you very much for your comment and suggestion to publish the manuscript.

Reviewer 3 Report

Epidermal Growth Factor Receptor (EGFR) is a transmembrane receptor tyrosine kinase that plays a crucial role in regulating cell growth and survival. Mutations or overexpression of EGFR are frequently observed in various cancers. Targeted therapies like EGFR inhibitors (e.g., Erlotinib, Gefitinib) have been developed to block EGFR signaling and have shown efficacy in specific patient populations. Targeted protein degradation has emerged as a promising strategy for cancer treatment, offering a potential alternative to traditional small molecule inhibitors. While small molecule inhibitors typically work by blocking the function of a protein, targeted protein degradation aims to remove the protein of interest from the cell by inducing its degradation. Furthermore, different approaches have been developed to target EGFR for degradation, primarily utilizing two main strategies: first, Proteolysis Targeting Chimeras (PROTACs), and second, Specific and Nongenetic Inhibitor of Apoptosis Protein (IAP)-dependent Protein Erasers (SNIPERs).

In the first part of the review, authors clearly described why chemotherapy with EGFR inhibitors is not an effective approach, due to emergence of resistance mutations. Next, authors talked about the therapeutic modality like proteolysis-targeted chimeras to target EGFR, could be an promising approach. However, as the approach is not tissue specific, it has off-target toxicity which largely limits the continued use of the targeted approach. Furthermore, AUTOphagy targeting chimera, an another therapeutic modulation could be an promising approach to treat cancer. However, authors mentioned that TPD is capable of slowing down the progression of cancer but not unable to completely impair the development and progression of cancer. The last part consist of new vision on cancer chemotherapy followed by a brief conclusion.

However, in the section ("New allosteric chemicals bind to EGFR and lead to cancer cell death"), authors discussed about an another effective alternative approach called allosteric autophagy (alloAUTO), which could potentially allow restoring the lost ability of cancer cells to die like normal cells after a limited number of generations. Here, I am offering some suggestions to strengthen the experimental section:

1) A densitometry graph for figure 4B could be helpful for quantitative       understanding of the effectivity of VM26 and VM25 compounds.

2) The decrease in the content of cyto-skeletal proteins β-actin and α-tubulin upon VM26 and VM25 treatments is an interesting observation, However, authors could add another ECM-cell adhesion marker in figure 4B.

3) Western-blotting for Bim level in VM26 and VM25 treated and untreated cell, or immunofluorescence staining using β-actin and α-tubulin could be valuable addition.

Additional suggestions:

1) All the proposed figures must be at higher resolution.

2) Authors could add brief discussion on SNIPERs approach for EGFR degradation.

3) Authors could consider making a comparison table, considering advantages and disadvantages among different existing targeted degradation approaches for EGFR, could be an excellent summary for the entire text. Addition of relevant references subsection would make the table informative.

Author Response

To Reviewer 3.

Dear Reviewer, we thank you for detailed analysis of our technology and appreciating it to improve targeted protein degradation in cancer.

Please find below our answers to your suggestions.

1) A densitometry graph for figure 4B could be helpful for quantitative       understanding of the effectivity of VM26 and VM25 compounds.

Response: Densitometric analysis of Fig. 4B was not performed in this experiment because it was not necessary. However, in other experiments, comparative densitometry was performed (see original article, ref. 135), which convincingly showed that VM26 is the strongest degrader of EGFR and other proteins in vitro. Considering this effect of VM26, it is considered the main FQTT molecule in our work.

2) The decrease in the content of cyto-skeletal proteins β-actin and α-tubulin upon VM26 and VM25 treatments is an interesting observation, However, authors could add another ECM-cell adhesion marker in figure 4B. 

Response: The review format limits the provision of detailed Figures. A more complete set of proteins, including caspases, has been studied in ref. 135. The discovery that cytoskeletal instability depends on Bim-mediated phosphorylation disfunction in VM26-treated cells was an important step in understanding the mechanism of cell detachment leading to cancer cell death. However, we agree with you and draw the reader's attention to other possibilities of provoking cell death using allosteric protein decomposers (see addition in lines 28-29 and lines 656-658).

3) Western-blotting for Bim level in VM26 and VM25 treated and untreated cell, or immunofluorescence staining using β-actin and α-tubulin could be valuable addition.

Response: It can be done. But we have already collected enough scientific information about the action of FQTT molecules. Please refer to the original article (ref. 135) for the proposed mechanism of action of the new drugs.

Additional suggestions:

1) All the proposed figures must be at higher resolution.

Response: Increased the resolution level of all shapes. We hope the quality has improved.

2) Authors could add brief discussion on SNIPERs approach for EGFR degradation.

Response: We have added ref. 129 for SNIPER. Unfortunately, we were unable to find an article on its use for EGFR degradation. For the same reason, other targeted approaches to protein degradation are not included in the review.

3) Authors could consider making a comparison table, considering advantages and disadvantages among different existing targeted degradation approaches for EGFR, could be an excellent summary for the entire text. Addition of relevant references subsection would make the table informative.

Response: Considering that the topic of directed protein degradation is developing rapidly and requires constant attention, it is not easy to present such a table. Therefore, we decided to invite the reader to take a look at the Creative Biolabs® website, which describes the advantages and disadvantages of the flagship PROTAC technology (lines 639-640, ref. 161 highlighted in red in chapter 7. Conclusions). We do not think that copying these explanations in our review can improve its quality. However, we mention some other disadvantages of targeting protein inhibition and degradation in cancer treatment.